# Effect of Soaking Conditions and Fuzzy Analytical Method for Producing the Quick-Cooking Black Jasmine Rice

**DOI:** 10.3390/molecules27113615

**Published:** 2022-06-04

**Authors:** Patareeya Lasunon, Nutchanat Phonkerd, Somprasong Pariwat, Nipaporn Sengkhamparn

**Affiliations:** Faculty of Interdisciplinary Studies, Khon Kaen University, Nong Khai Campus, Nong Khai 43000, Thailand; lpatar@kku.ac.th (P.L.); nutchanat@kku.ac.th (N.P.); somppa@kku.ac.th (S.P.)

**Keywords:** quick-cooking black jasmine rice, rehydration capacity, phenolic compound, anthocyanin, fuzzy analytical method

## Abstract

The quick-cooking rice product is an interesting product for the market which is easy to cook, with good sensorial qualities and health benefits. This work aimed to study the effect of the soaking conditions, namely baking powder concentration (0.1, 0.2, or 0.3%), soaking temperature (room temperature, 50 or 60 °C), and soaking time (10, 20, or 30 min), in order to improve the physical properties and also the sensory characteristics, with high bioactive compound content, of Quick-Cooking Black Jasmine Rice (QBJR). The physical properties of the final product, namely the rehydration capacity, morphology, and texture, were observed. Moreover, the total phenolic, total flavonoid, and total anthocyanin were determined. The results showed that the samples with a high baking powder concentrations soaked at high temperatures for longer time affect the low rehydration capacity with a high hardness value and a decreased bioactive compound content. In addition, the sensory score including softener, flavor, and overall acceptance were lower score. Moreover, to determine the best soaking condition with complex data, the Fuzzy Analytical Method (FAM) was performed by an online FAM program. The results showed that soaking at room temperature for 30 min in 0.1% of baking powder showed the highest overall performance index of 6.52.

## 1. Introduction

Nowadays, the lifestyle of people has changed, especially regarding food consumption behavior. People love to consume fast or quick-cooking foods [1]. Therefore, the quick-cooking rice consumption has increased [1]. Quick-cooking rice is partially cooked rice that is then dried. The production condition affects its product quality. Oikonomopoulou, Krokida, and Karathanos [2] reported that the cooking methods affect rice’s structural properties including its porosity, bulk density, and the true density of the final sample in which the vacuum employed during freeze-drying increased the bulk density but decreased the porosity of the grain. Song et al. [3] studied the effects of baking powder concentration, cooking temperature, and freezing temperature on the quality of freeze-dried cooked rice. They found that the higher baking powder concentrations increase the rehydration ratios of the final sample. Bui et al. [4] compared the rice type and processing conditions on the functional properties of freeze-dried rice and found that par-boiled rice was more suitable for the freeze-drying process. Zhu et al. [5] studied the effect of the soaking temperature on the quality of cooked brown rice and found that the higher temperature (60 and 70 °C) enhanced the penetration of water to the grain leading to the springiness of the grain. However, the effect of the baking powder concentration together with soaking temperature and soaking time on quick-cooking black jasmine rice has never been reported before.

Black rice has higher nutritional and bioactive components than white rice. It contains both macronutrients, namely essential amino acids, lipids, and dietary fiber, as well as micronutrients including vitamins (B complex, A and E), some minerals (K, Fe, Zn, Cu, Mg, Mn, and P), as well as bioactive components [6]. The phenolic compounds contain one or more hydroxyl groups on an aromatic ring [6]. Duyi et al. [7] reported that the flavonoids in rice grains were O- or C-glycosides. The main pigment in black rice is anthocyanins, which is a water-soluble pigment delivering a reddish to purple color. Anthocyanins are polyhydroxylated and or methoxylated heterosides [7] and petunidin or malvidin aglycones are more stable than pelargonidin, cyanidin, or delphinidin aglycones [8]. Moreover, the amount of glycosylation increases its stability [8]. The majority, specifically 88%, of total anthocyanin in black rice is cyanidin-3-glucoside, Cy3G [9]. The bioactive compound is generally unstable in a high pH environment and also in thermal processing.

The mathematical techniques used in research, namely the fuzzy analytical method (FAM), are decision-making methods used for efficiency. By using this method, the data was transferred to a fuzzy scale, a fuzzy grade, and the overall performance index was calculated. This method has been used for complex decision-making by using overall performance index values. FAM has been reported to be used for the assessment for sensory evaluation [10,11,12] for extraction conditions [13,14,15]. So far, the FAM has been applied for the evaluation of conditions with seven criteria [16], in which all criteria contained equal amount of data. However, no report has been found for applying FAM on more than seven criteria, and for each criterion containing unequal data.

Therefore, in this study aimed to study the effects of baking powder concentration, soaking temperature, and soaking time on the physical properties, bioactive compound contents, as well as the sensory characteristics. Moreover, for the complex data, the decision for the best soaking condition has been evaluated using the Fuzzy analytical method.

## 2. Results

### 2.1. Physical Properties of Quick Black Jasmine Rice

The physical properties and also microstructure of cooked rice has been improved by using baking powder [3] and also by adjusting the soaking temperature and soaking time [5]. The Quick Black Jasmine Rice were differently pretreated with baking powder solutions of 0.1, 0.3, or 0.5% at room temperature, 50, and 60 °C for 10, 20, and 30 min. The physical properties of the samples including the rehydration capacity, morphology, and texture of the samples were observed. The rehydration capacity of QBJR samples with reference to the amount of absorption water are shown in Figure 1.

The results showed that the concentration of baking powder, soaking temperature, as well as time affected the rehydration capacity. A higher soaking temperature and a longer soaking time decreased the rehydration capacity of QBJR samples. Moreover, the concentration of baking powder had not significantly affected the rehydration capacity of sample which soaked at a high temperature. This probably due to the different porosities of the samples during drying. Therefore, the morphologies of the sample were also identified and the results are shown in Figure 2.

Figure 2 shows that the higher soaking temperature resulted in a higher small pore, while at high baking powder concentrations, a hollow channel could be found. However, the lower soaking temperature caused a bigger hollow channel in the center of the grain. This appearance was related to the higher rehydration capacity of the samples with soaking at room temperature. Moreover, the soaking time did not significantly affect the morphology of the sample grains.

After rehydration, the texture of QBJR samples were determined and the results are shown in Figure 3. The results revealed that the hardness of the sample was interfered with by three factors of soaking processing. At higher soaking temperatures, more hardness was found in the sample. On the contrary, the springiness of the sample was affected by the concentration of baking powder. However, the baking powder concentration of 0.3% gained the highest values of hardness and springiness.

### 2.2. Bioactive Compound Content of Quick Black Jasmine Rice

Black rice is rich in nutrients such as proteins, vitamins, phenolic compounds, and also anthocyanin, making the rice have a black color. However, this bioactive compound has been affected by the processing conditions. For producing the high quality Quick Black Jasmine Rice, therefore, the total phenolic (TPC), total flavonoid (TFC), as well as the total anthocyanin (TAC) content of each QBJR sample were determined. The results are shown in Figure 4. Therefore, the total phenolic (TPC), total flavonoid (TFC), as well as total anthocyanin (TAC) content of each QBJR sample were determined. The results are shown in Figure 4.

The results exhibited that the three factors including baking powder concentration, soaking temperature, and soaking time have affected the bioactive compound in the QBJR sample. It is found that the higher baking powder concentration decreased all bioactive compounds; moreover, the higher soaking temperature and longer soaking time enhanced the leaching of all bioactive compounds.

### 2.3. Sensory Evaluation

To evaluate the best condition of soaking, the sensory characteristics of each QBJR was determined. The 30 Thai students in Faculty of Interdisciplinary Studies, Khon Kaen University, Thailand, evaluated the rehydrated QBJR sample with regards to three important attributes including flavor, softener, and overall acceptance by using a nine-point hedonic scale, in which very poor (weak) corresponded to a score of 1.0, while very good (intense) corresponded to a score of 9.0. The results are shown in Figure 5. The results pointed that the flavor score was increase when high baking powder concentrations were used. Moreover, the softener has been affected by the baking powder concentration, soaking temperature, and soaking time. It was found that a high softener score was identified by using the treatment of a higher baking powder concentration, a longer soaking time, and particularly at a low soaking temperature. Moreover, the overall acceptance score was also in agreement with the softener score; when a high baking powder concentration at a longer time was used, the high overall acceptance score was found.

### 2.4. Fuzzy Assessment Method

In order to evaluate the best condition for the QBJR samples with the high physical properties, a high bioactive compound content, and a high sensory score, the fuzzy assessment was performed. The nine sub-criteria and their weights were used to examine the following details. For the physical properties, the data regarding rehydration capacity, hardness, and springiness were used. Each datum of the physical properties was weighted to 15, 10, and 5% for rehydration capacity, hardness, and springiness, respectively. Moreover, the weight of the bioactive compound was 10, 10, and 20% for TPC, TFC, and TAC, respectively. In the meanwhile, the weight of each attribute of sensory score was equal to 10%. The overall performance index of 27, indicating the processing condition, is shown in Figure 6. The best condition was the soaking the BJR in 0.1% of baking powder at room temperature for 30 min with the highest overall performance index of 6.52.

## 3. Discussion

### 3.1. Effect of Processing Condition on Physical Properties of QBJR

The dried rice quality has been affected by the processing condition which caused the physical properties of the product such as rehydration and microstructure to change, and also caused the bioactive compound concentrations to vary. Song et al. [3] stated that the microstructure of cooked rice during drying affected the rehydration properties, which were improved and resulted in a high quality of the final product. Therefore, the processing of QBJR, namely baking powder concentration, soaking temperature, and soaking time, on rehydration capacity, microstructure, and also the texture of final product were studied.

The rehydration capacity exhibited the capacity of the dried sample to absorb and retain the water in the sample. This is a one of many important parameters for dried samples, which depends on many factors. The results showed that baking powder concentration, soaking temperature, and soaking time has affected the rehydration capacity. At room temperature being used as the soaking temperature, the increased baking powder concentration increased the rehydration capacity, however, the baking powder concentration of 0.3 and 0.5% was not different. This was in accordance with Song et al. [3] who stated that the higher concentration of baking powder used, the higher the rehydration ratio of rice. This can be explained by the gas production of baking powder causing the space in the final QBJR structure after the drying process, which improves the rehydration capacity of the samples. Moreover, the rehydration capacity of the sample was not differed at baking powder concentrations of 0.3% at a soaking temperature of room temperature and that of 50 °C. However, at the soaking temperature of 60 °C, the increased baking powder concentration did not affect the rehydration capacity of QBJR samples. This was probably due to a high concentration; the generated gas may be evaporated during soaking processing.

Considering the effect of soaking time, the results exhibited that the longer soaking time (30 min) caused the decrease of rehydration capacity especially at high soaking temperature, but no effect found for a soaking time of 10 and 20 min. This probably can be explained by the maximum swelling of starch granules at high temperatures, and probably the breakdown of starch granules, which may affect the capacity of the samples to retain the absorbed water during the rehydration process. Moreover, at high soaking temperatures, the starch or soluble material would be subjected to increased leaching from the kernel [17]. This would lead to the lower rehydration capacity of samples at a high soaking temperature for a longer time.

To explain the processing factors on the rehydration capacity of rice samples, the morphology of the samples was observed. Obviously, the morphology of the samples was not affected by the soaking time. Therefore, the morphology is shown for the same soaking time of 10 min (Figure 2). The results revealed that the morphology of the samples, which soaked at 50 and 60 °C, had large amounts of the small pore structure, a honeycomb pattern, especially at soaking temperature of 60 °C, and also some large-pore structures. This appearance was also found in research work conducted by Zui et al. [5] who found that for the rice soaked at 50 −70 °C, the honeycomb pattern was observed and some cracks were also found. This can be explained by the loosening of granules and the swelling of the starch granules. This would be in agreement with the more-swelled starch granules, and may be some broken down starch granules at a high temperature of soaking. On the contrary, for the morphology of the samples which soaked at room temperature, the large pore structure was observed and we did not observe the small pore structure. This large pore structure seems to affect the high rehydration capacity. It can be pointed out that the processing condition caused the difference in sample morphology which is in accordance with Zui et al. [5].

The texture properties of rehydrated samples including hardness and springiness is also one important factor for studying the eating quality of cooked rice. The bite resistance of samples refers to their hardness value, meaning a firm structure of rice. The results (Figure 3) exhibited that at a low soaking temperature and increased concentrations of baking powder, the hardness was decreased. This was related to the rehydration capacity, for which lower hydration resulting in a high hardness value. Moreover, at high soaking temperatures the hardness of samples was higher than at low soaking temperatures for every soaking time. This would be the changes of the starch granules which could be broken down during the soaking procedure. Consequently, the sample could not store the absorbed water hence the grain has high hardness. Concerning the springiness of the sample, which refers to the deforming of the clump in the mouth, it is found that with the increase of baking powder concentration from 0.1 to 0.3%, the springiness was increased, but the springiness of the samples soaking at 0.1% of baking powder concentration was similar to the 0.5% concentration. Moreover, the effect of soaking temperature and time did not affect the springiness. Song et al. [3] found that when they compared the quick rice sample that was soaked in baking powder to the unsoaked sample, the springiness value was lower in the latter sample. This also pointed out that the gas production from soaking in baking powder solution would improve the springiness. From the results, it showed that the baking powder concentration of 0.3% gained the highest values of hardness and springiness.

### 3.2. Effect of Processing Condition on Bioactive Compound Content

The black rice is rich in nutrients and also bioactive components, for example, essential amino acids, lipids, dietary fiber, vitamins (B complex, A and E), some minerals (K, Fe, Zn, Cu, Mg, Mn, and P), anthocyanins, and phenolic compounds [6]. This bioactive compound can be destroyed during the rice processing. In addition, the effect of quick rice production on bioactive compound concentrations has not been demonstrated yet. Therefore, the bioactive compound of QBJR were determined and discussed.

This research has studied three processing factors on the total phenolic (TPC), total flavonoid (TFC), as well as total anthocyanin (TAC) content as shown in Figure 4. The results exhibited that the higher baking powder concentration and soaking temperature for a longer time declined the bioactive compound content in QBJR. This can be explained by the pH-sensitive bioactive compound, for which the baking powder conferred a higher pH to the soaking water and resulted in the unstable phenolic, flavonoid, and anthocyanin contents in the QBJR. Moreover, these compounds are heat-sensitive and water soluble, hence the high soaking temperature and longer soaking time resulted in the destroying and leaching of all bioactive compound to the soaking water.

Phenolic compounds, divided into two groups, consists of (1) flavonoids and phenolic acids and (2) coumarins, for which their structures contain one or more hydroxyl groups on an aromatic ring [6]. The flavonoids in rice grains have been reported to be O- or C-glycosides [7]. The phenolic compounds were unstable also in the high-pH environment. Friedman and Jürgens [18] reported that polyphenolic compounds are damaged when exposed to a high pH. Moreover, the thermal processing also affect the phenolic composition by chemical and physical reactions. Duodu [19] stated that during thermal processing the matrix-bound phenolics can be released, polymerized, and/or be thermally degraded. Lang, et al. [20] reported that during different drying temperature of black rice, the free phenolic compounds can be complexed with some structural components, such as proteins or fibers at drying temperature of 60 and 80 °C and phenolics can be degraded at a drying temperature of 100 °C.

Moreover, the black rice also contained a high amount of anthocyanin, which is mainly found as cyanidin-3-glucoside, Cy3G [9]. Anthocyanin is a water-soluble natural food pigment and the stability of anthocyanin depends on its structure, in which petunidin or malvidin aglycones are more stable than pelargonidin, cyanidin, or delphinidin aglycones [8]. Moreover, the increase in glycosylation will improve its stability [8]. Anthocyanins can be degraded with an alkaline pH. Sui, Ding and Zhou [21] stated that they were increasingly degraded at the pH level of above 6.0 and under thermal treatment. Cabrita, Fossen, and Andersen [22] reported that cyanidin3-glucoside was highly degraded at a pH about 5.0–6.0. In our study, the baking powder solution was found to have a pH of about 7.0; this pH values would affect the anthocyanin content. However, the objective of this work was to study the best soaking conditions to produce the Quick Black Jasmine Rice in order to make it easier and faster to cook, and from our results it showed that the final product retained the high bioactive compound.

Moreover, one of the main factors that reduces the stability of Cy3G is thermal processing [23]. Dong et al. [23] reported that from thermogravimetric analysis showed that the Cy3Gs are thermally degraded in three steps, in which the first step occurred at 120–210 °C. Moreover, they also stated that the anthocyanin-rich foods could be conventionally cooked, for example through steaming, boiling, and microwave heating, in which the cooking temperatures would be reached about 50–150 °C. Therefore, in our experiment the soaking temperature did not destroy the anthocyanin, especially Cy3G, even though some anthocyanin compound may be transformed. However, to look into this more deeply, the anthocyanin profile of the final sample could be determined.

Even though the soaking process could improve the physical properties of QBJR, however, some bioactive compounds would be unstable during processing. Therefore, this point could be concerned in order to produce the quick rice product especially for black rice, which are rich in bioactive compounds.

### 3.3. Effect of Processing Condition on Sensory Evaluation

The sensory characteristics of the QBJR were evaluated by 30 panelists, Thai students who studied in Bachelor of Science Program in Food Technology and Innovation, Faculty of Interdisciplinary Studies, Khon Kaen University, Thailand, for three attributes: softener, flavor, and overall acceptance. The sensory score of the samples is shown in Figure 5. The three processing variables were evaluated on terms of their effects on the sensory score of the final samples. The results showed that the concentration of baking powder influenced the flavor score, in which for higher concentrations, the flavor score was increased. Even though the baking powder would have some aftertaste effect, however, in our study we found that the panelists did not seem too unappreciative of the flavor of the final sample which soaked at a high baking powder concentration. Concerning the softener score, the results exhibited that the baking powder concentration, soaking temperature, as well as the soaking time affected the softener score of samples. The increase of baking powder concentration resulted in an increase of softener score especially at a low soaking temperature; however at high temperatures, the concentrations of 0.3 and 0.5% was not differ. This result was in accordance with the rehydration capacity of the sample in which for the high rehydration capacity, the high softener score was observed. Furthermore, the soaking time at a low temperature influenced the softener score, in which for the longer soaking time, a higher softener score was found. These were in agreement with the hardness value delivered from the texture analyzer profile. In addition, the overall acceptance score of QBJR samples increased when higher baking powder concentrations were used. Meanwhile, for the longer soaking time, the overall acceptance score was increased. This result was in agreement with hardness values as well as the hardness score of QBJR samples. Consequently, all sensory evaluation data demonstrated that the soaking procedure could affect the sensory score and the physical properties of the samples, which were also related to the sensory score.

The evaluation for the best condition for QBJR production concerning the physical properties, sensory characteristics, as well as the amounts of bioactive compounds were performed by the Fuzzy analytical method. This method is discussed in the following section.

### 3.4. The Evaluation the Best Processing Condition

The FAM is a mathematical technique and has been performed for decision making. This method transfers all data to a fuzzy scale, fuzzy grade, and to the overall performance index. Moreover, the application of FAM on more than seven criteria for which each criterion contains unequal data has never been reported before. In this work, the main criterion was divided into three criteria, namely physical properties, bioactive compound content, and sensory score with the wight of 30:40:30, respectively. In the physical property criterion it was separated into three sub-criteria, which were the rehydration capacity, hardness values, and springiness values. Conferring these criteria to the overall physical properties, the weight of each sub-criteria was 15, 10, and 5 for the rehydration capacity, hardness values, and springiness values, respectively. Considering the importance and specificity of bioactive compounds in black rice, the total anthocyanin content was the main sub-criteria of this criteria with the weight of 20, while the weight of the total phenolic content and total flavonoid content were equal, being 10. For the sensory score data, the sub-criteria were flavor, softener, and overall acceptance score with the same weight. Furthermore, in each QBJR sample, the data of each sub-criteria in terms of physical properties as well as bioactive compound criteria consisted of three sets of experimental data, while the data of sensory score criteria consisted of 30 sets of experimental data. According to each sub-criterion, it consisted of unequal data, therefore all 108 pieces of data of each sample were translated into the fuzzy scale that was designed (0, 10), for which the number 10 has been determined as perfect performance, as the following equation:(1)$$\mathrm{Fuzzy}\;\mathrm{score}=\left(\frac{S_{i}-a}{b-a}\right)\times \left(10-0\right)$$

For all data $S_{i}\in \left[a,b\right]$, where a is the maximum and b is the minimum of each criterion from all sample.. Except for hardness criteria, which has to be transferred in reverse, in which the maximum scale is not perfect, the fuzzy scale of the hardness criteria can be translated as the following equation:(2)$$\mathrm{Fuzzy}\;\mathrm{score}=\left(\frac{b-S_{i}}{b-a}\right)\times \left(10-0\right)$$

So, each data in the different boundary were transformed in the same range [0, 10], for which this range was used to define the fuzzy performance grade sets by using the triangular fuzzy numbers, as seen in Figure 7.

Next, the fuzzy score of each sub-criteria were calculated into the fuzzy performance grade sets *A*, *B*,…, *F* (ranging from the best to the worst), which was then used to integrate with each relative weight, which was called the index score are is denoted by $I={\left[I_{A},I_{B},I_{C},I_{D},I_{E},I_{F}\right]}^{T}$. Then, all 108 index scores were combined to calculate the overall index. Therefore, the obtained overall index was calculated from the same fuzzy performance grade set. The overall index of each sample (Figure 6) exhibited that the soaking in 0.1% of baking powder at room temperature for 30 min was the best soaking condition for producing the QBJR. This condition showed high physical quality with a high amount of bioactive compounds, as well as a high sensory score. This shows that FAM can be used as an efficiency evaluation method for the discission of the best condition in this present work, especially when many criterions and sub-criterions were concerned. Moreover, it is an effective tool for unequal data and also some data have to be considered in a reverse response.

## 4. Materials and Methods

### 4.1. Materials

The black jasmine rice was collected from Nong Khai Rice Research Center. The rice was an organic rice which was grown in the temperature of 30–35 °C during January until March 2021.

Folin-Ciocalteu reagent (#03870) was purchased from LOBA CHEMIE PVT. LTD. (Wodehouse Road, Colaba, Mumbai, INDIA). Sodium carbonate anhydrous (AJA463-500G) was purchased from AJAX Finechem, (Bay Rd., Taren Point, Australia). Ethanol absolute AR grade (#AR1380-P), methanol AR grade (#AR1115-P) and hydrochloric acid 37% (#AR1107-G) were purchased from RCI Labscan Limited (Pathum Wan, Bangkok, Thailand). Quercetin (>95% HPLC grade, #Q4951) and gallic acid (#G7384) were purchased from Sigma-Aldrich (St. Louis, MO, USA). Aluminium nitrate 9H2O (#A4018) was purchased from Qrec, (Auckland, New Zealand). Sodium nitrite AR grade (#492) was purchased from KemAus (Bangchak, Phakanong, Bangkok, Thailand).

### 4.2. Quick-Cooking Black Jasmine (QBJR) Preparation

The black jasmine rice (BJR) was rinsed with water and then soaked in the different concentrations of baking powder solution (0.1, 0.3 and 0.5%) and different temperatures (room temperature, 50, and 60 °C) for 10, 20, and 30 min, and the ratio of rice to solution was 1 to 25 *w*/*v*. After that, the BJR was cooked in a rice cooker at cooking temperatures of about 100 °C. The cooked BJR was kept at 4–5 °C for 15 h and then was dried at 50 °C for 4 h and 95 °C for 2 h. The Quick Cooking Black Jasmine (QBJR) sample from each condition was kept in a plastic bag and stored at room temperature before further analysis. The schematic diagram of Quick Cooking Black Jasmine production in this work is shown in Figure 8.

### 4.3. Physical Properties of QBJR

#### 4.3.1. The Rehydration Capacity

The rehydration capacity of QBJR was determined according to Song et al. [3] with some modifications. Briefly, 30 g of QBJR was added with 300 mL of hot water (90 °C) then the QBJR was allowed to rehydrate for 15 min before the excess water was drained. The rehydrated QBJR was weighed. The rehydration capacity of the sample was calculated as follows:Rehydration capacity % = 100 × (W_2_ − W_1_)/W_1_(3)
where W_2_ is the weight of the rehydrated sample, and W_1_ is the weight of the dry sample [24].

#### 4.3.2. Scanning Electron Microscopy

The morphology of QBJR before rehydration was examined using a scanning electron microscope (SEM) (JEOL JSM-6010LV, Tokyo, Japan). The cross sections of each treatment were then placed on a double adhesive tape, affixed on the surface of a metal stub, and coated with gold.

#### 4.3.3. Texture Profile Analysis

The texture of QBJR after rehydration was measured using a texture analyzer (Stable Micro Systems/TA-XT plus, Godalming, Surrey GU7 1YL, UK) with a 50 cm cylindrical probe according to Bui et al. [4] with some modifications. The test sample was tested under a 500 N load cell, a compression speed of 5 mm/sec, a return seed of 15 mm/sec, a contact force of 5 g, and a maximum compression of 75%. The hardness, adhesiveness, stickiness, and chewiness were recorded.

### 4.4. Bioactive Compound Content

#### 4.4.1. Bioactive Compound Extraction

The bioactive compounds in QBJR were extracted according to Meng et al. [25] with minor modifications. The QBJR was ground into powder and about 0.5 g was extracted with 80% methanol under a sonication bath at room temperature for 2 h. Then, the sample was centrifuged at 4000 g for 15 min and the supernatant was collected for total phenolic content and total flavonoid content determination.

#### 4.4.2. Total Phenolic Content Determination

The total phenolic content of QBJR was determined using the Folin–Ciocalteu method according to Meng et al. [25]. An amount of 5 mL of distilled water and 0.5 mL of Folin–Ciocalteu reagent were added to 0.2 mL of sample extract and the mixture was allowed to stand for 5 min. Then, 1.5 mL of sodium carbonate (75 g/L) was added to the mixture and let to stand for 1.5 h in the dark. The absorbance of the mixture at 725 nm was recorded. The TPC was reported as mg of gallic acid equivalent (GAE) per gram of samples (mg GAE/100 g sample) with a standard curve equation of y = 0.471x − 0.0146 (R^2^ = 0.990) where y is the absorbance of the mixture at 725 nm and x is the gallic acid concentration.

#### 4.4.3. Total Flavonoid Content (TFC) Determination

The total flavonoid content of QBJR was determined according to Meng et al. [25] with some modifications. An amount of 2 mL of distilled water and 0.15 mL of 5% sodium nitrite were added to 0.3 mL of sample extract and then let to stand for 5 min. Then, 0.15 mL of aluminum nitrate was added and let to stand for 5 min, and 1 mL of 1 M sodium hydroxide was added. The absorbance of the mixture at 420 nm was recorded. The TFC was reported as milligrams of quercetin acid equivalent (QAE) per gram of samples (mg QAE/100 g sample) with a standard curve equation of y = 0.2408x + 0.1632 (R^2^ = 0.995) when y is the absorbance of the mixture at 725 nm and x is the quercetin concentration.

#### 4.4.4. Total Anthocyanin Content (TAC) Determination

The total anthocyanin content (TAC) of QBJR was determined according to Chirawat et al. [26]. An amount of 10 g of QBJR was macerated in 10 mL of 1% hydrochloric acid in ethanol at 60 °C for 4 h. The mixture was then filtrated. Then, 0.2 mL of supernatant was diluted with 1% hydrochloric acid in ethanol to 10 mL of total volume. The absorbance of the mixture at 535 nm was recorded. TAC was calculated as follows:TAC (mg/10 mL) = [(A535 × A × 100)/(0.2 × B)]/98.2(4)
where A was the total volume of the extract (mL) and B was weight of QBJR (mg).

#### 4.4.5. Sensory Evaluation

The sensory acceptability of the rehydrated QBJR samples was tested About 3 g of each rehydrated QBJR sample was served to 30 Thai panelists, which consisted of 9 males and 21 females in the age range of 18–25 years. All panelists studied in the Bachelor of Science Program in Food Technology and Innovation, Faculty of Interdisciplinary Studies, Khon Kaen University, Thailand. The three attributes of hardness, flavor, and overall acceptance were evaluated using a nine-point hedonic scale.

#### 4.4.6. Statistical Analysis

The experimental design was done using factorials in CRD (3 × 3 × 3) all the data were determined in triplicate. The effect of each factor on response data was calculated at the significance level of 0.05.

#### 4.4.7. Fuzzy Analytical Method (FAM)

The FAM is the numerical analysis methodology used to calculate the overall index of all data from many decision criteria [13]. The overall index is calculated by integrating the fuzzy performance grade sets with their relative weights. The best processing condition (out of 27 conditions) of Quick Cooking Black Jasmine rice was evaluated from 3 criteria, including physical properties, bioactive compound contents, and sensory score, which were used to exam the condition with the weight of 30:40:30, respectively. The physical property criterion consisted of a rehydration capacity, hardness values, and springiness values with the weights of 15, 10, and 5, respectively. The bioactive compound content criteria consisted of total phenolic content, total flavonoid content, and total anthocyanin content with the weights of 10, 10 and 20, respectively. Meanwhile, the sensory score data consisted of flavor, softener, and overall acceptance score with the same weight of 10 per each. Each criterion consisted of 3 sub-criteria, so there were 9 criteria to consider. Three replication for performed for all the data, except the sensory score, which contained 30 datapoints per sample. Due to the difference of data in each criterion and also the big obtained data (108 data per one sample), the overall performance index was calculated by an online FAM program, which was developed by Lasunon, et al.

## 5. Conclusions

The Quick-Cooking Black Jasmine rice were tested by 27 different conditions, which varied the baking powder concentration, soaking temperature, and soaking time. The physical properties, chemical properties, as well as the sensory properties were determined. The results showed that the increase of baking powder concentration soaked for a shorter time, especially at room temperature, increased the rehydration capacity and showed a lower hardness value. Moreover, the morphology of this sample showed a large pore structure which affected the capacity of the sample to absorb water. Furthermore, the bioactive compound levels decreased when a higher concentration of baking powder was used at a high temperature of soaking for longer time. However, the final product still contained a high amount of bioactive compounds. Concerning the sensory characteristics, it was found that the higher concentration of baking powder did not affect the flavor score but decreased the softener score, especially at a low temperature of soaking. In addition, the overall acceptance score was related to the softener score. The best soaking condition for high physical properties, bioactive compound content, and high acceptance were evaluated by FAM, for which all data were calculated as the same size set. This found that FAM was a beneficial evaluation method for decision-making regarding complex data with many criterions to be concerned with.

## Figures and Tables

**Figure 1 molecules-27-03615-f001:**
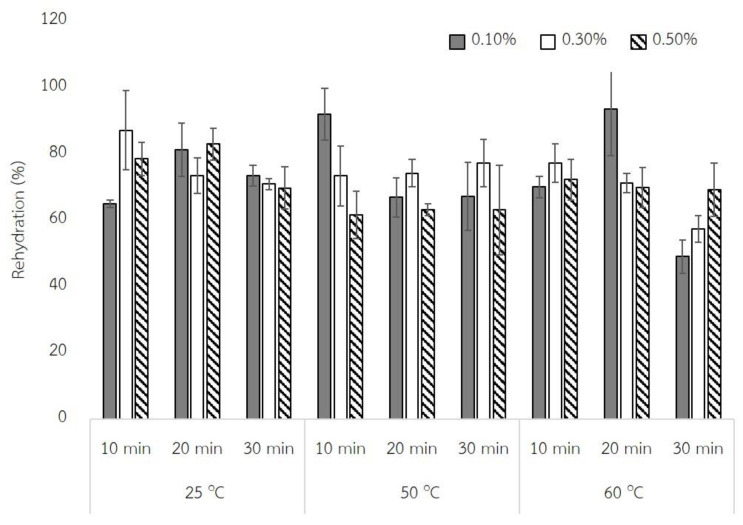
Rehydration capacity of each QBJR sample.

**Figure 2 molecules-27-03615-f002:**
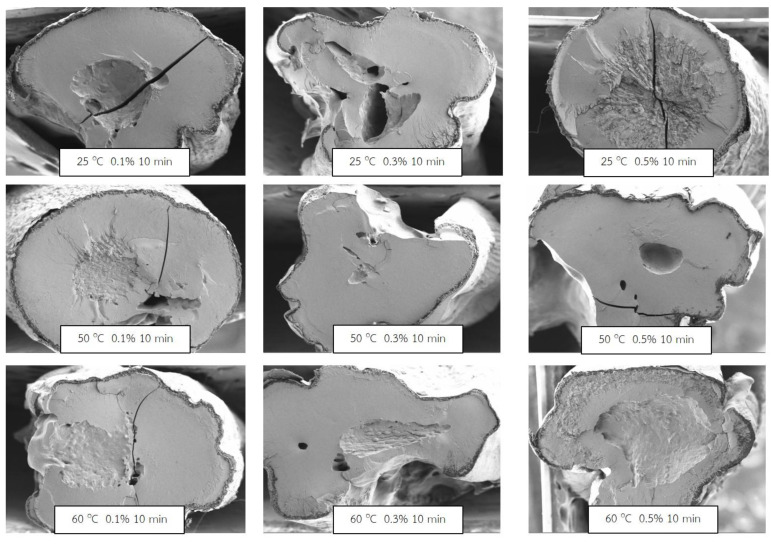
SEM morphology of QBJR sample at the same soaking time of 10 min.

**Figure 3 molecules-27-03615-f003:**
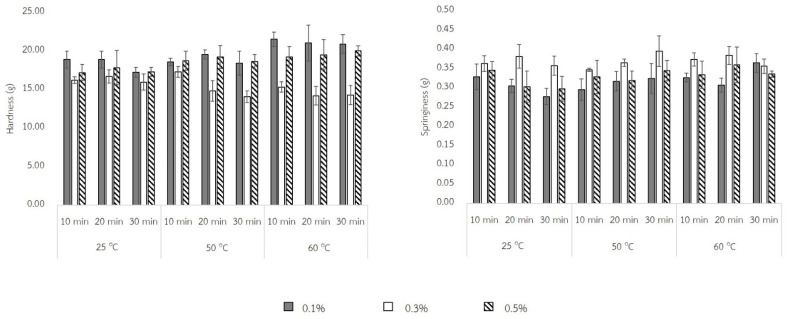
The hardness and springiness of each QBJR sample.

**Figure 4 molecules-27-03615-f004:**
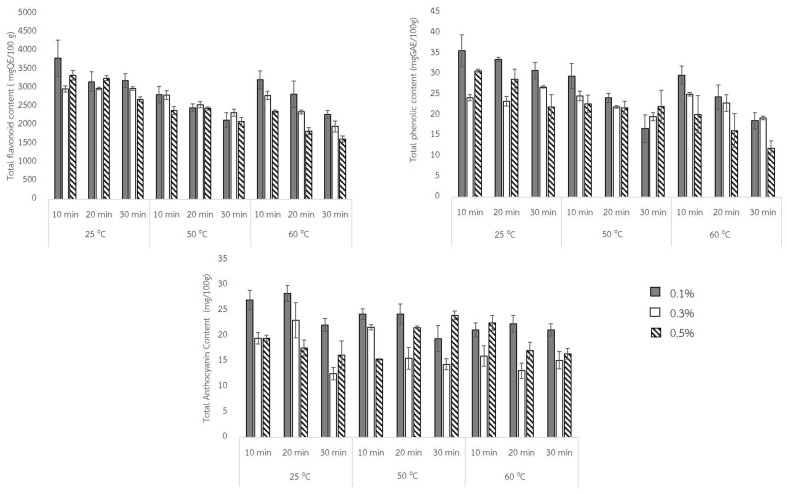
The total phenolic (TPC), total flavonoid (TFC), as well as total anthocyanin (TAC) content of each QBJR sample.

**Figure 5 molecules-27-03615-f005:**
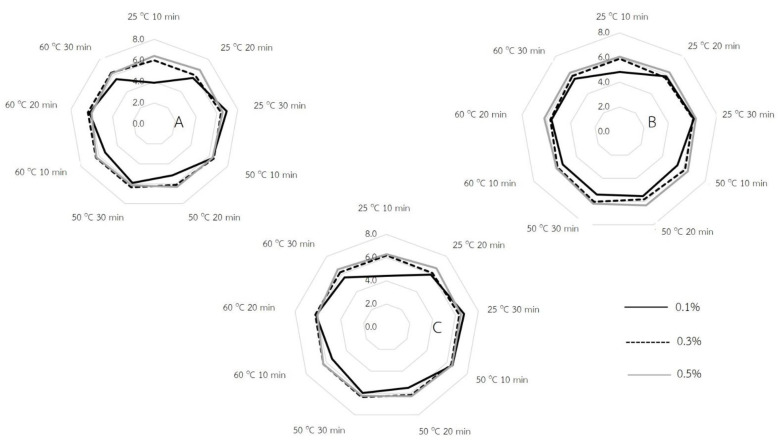
The sensory score of each QBJR sample. (**A**) softener, (**B**) flavor and (**C**) overall acceptance.

**Figure 6 molecules-27-03615-f006:**
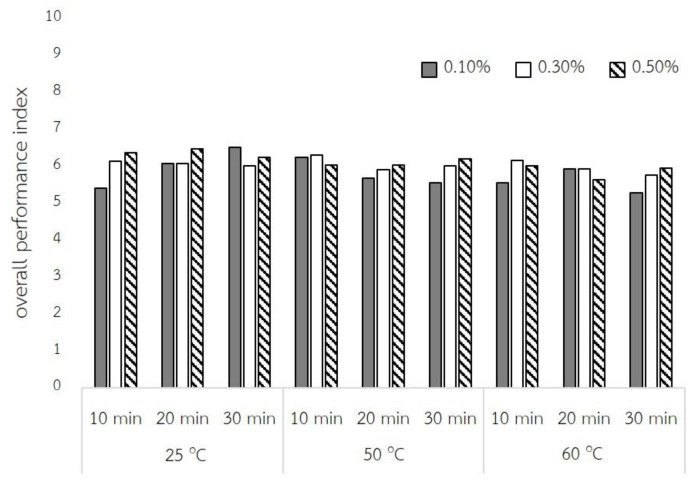
The overall performance index of each QBJR sample.

**Figure 7 molecules-27-03615-f007:**
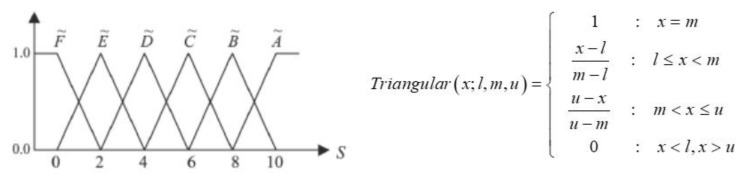
The triangular fuzzy grade and its membership function.

**Figure 8 molecules-27-03615-f008:**
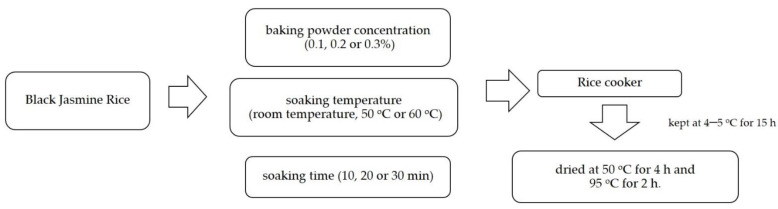
The schematic diagram of Quick Cooking Black Jasmine production in this work.

## Data Availability

Data sharing is not applicable to this article.

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
