# Peer review of "Effect of Soaking Conditions and Fuzzy Analytical Method for Producing the Quick-Cooking Black Jasmine Rice"

_molecules, 2022, doi:10.3390/molecules27113615_

Round 1
Reviewer 1 Report
The importance of study the effect of cooking, processed food and complex food matrices on food quality should be better marked and related references added such as:
Durazzo et al. Nutritional composition and antioxidant properties of traditional Italian dishes. Food Chem. 2017;218:70-77. doi:10.1016/j.foodchem.2016.08.120
The aim and the novelty character of paper should be better marked.
Introductory lines should be added in section Results to better introduce and describe the different type of results.
Data in Figure 3 and Figure 4 should be better described in the text.
The results reporting the sensory evaluation should be better described and discussed.
Additional information on samples and sampling should be given.
Author Response
Answer the Reviewer 1
(1) The importance of study the effect of cooking, processed food and complex food matrices on food quality should be better marked and related references added. The aim and the novelty character of paper should be better marked.
Answer The introduction according to the research gap has been added in the line 40-42, and 59-67
(2) Introductory lines should be added in section Results to better introduce and describe the different type of results.
Answer The introductory lines have been added into the Results such as in the line.70-71, 103-108, 119-124, 133-135, 294-310.
(3) Data in Figure 3 and Figure 4 should be better described in the text.
Answer Data in Figure 3 and 4 has been discussed in the text in the line of 198, 224.
(4) The results reporting the sensory evaluation should be better described and discussed.
Answer The sensory evaluation result has been more described in text in the line 266-268.
(5) Additional information on samples and sampling should be given.
Answer The information about samples such as information about the material (black jasmine rice), the sensory evaluation, texture profile analysis has been added to the part of Material and Method.
Reviewer 2 Report
The manuscript molecules-1730291 reported the Production of Quick Cooking Black Jasmine Rice: Effect of Soaking Condition and Fuzzy Analytical Method. The results bring us some new information about the influence of the pretreatment process of Quick Cooking black jasmine rice on its quality, and establish a method to analyze the best conditions. However, this article lacks innovation, major revisions are necessary. The following questions need to be answered.
(1) The author does not highlight the purpose and innovation of his work in the introduction. The author needs to revise the introduction.
(2) The room temperature in soaking time is not precise and needs to be expressed by definite temperature. And why choose 50 and 60℃ as soaking time?
(3) SEM does not indicate the magnification factor, and the author does not specify other treatment conditions (Soaking concentration and time). The authors need to display the SEM of the samples with different treatments.
(4) What is the significance of the Fuzzy Analytical Method establishment for this work?
(5) The article did not specify the gender or age composition of the sensory evaluation panel. The sensory evaluation data is suggested to be represented in the form of radar maps.
(6) Complete the error bars in Fig. 4.
Author Response
Answer the Reviewer 2
(1) The author does not highlight the purpose and innovation of his work in the introduction. The author needs to revise the introduction.
Answer The introduction according to the research gap has been added in the line 40-42, and 59-67
(2) The room temperature in soaking time is not precise and needs to be expressed by definite temperature. And why choose 50 and 60℃ as soaking time?
Answer the definite of room temperature was about 30oC and this information has been added in the material and method part. Moreover, according to Zhu et al. [5] found that the higher temperature (50, 60 and 70oC) enhanced the penetration of water to the grain leading to the springiness of grain, moreover, the gelatinization temperature of black jasmine rice was about 70oC (preliminary test). Therefore, this research conducted at the temperature of 50 and 60℃.
(3) SEM does not indicate the magnification factor, and the author does not specify other treatment conditions (Soaking concentration and time). The authors need to display the SEM of the samples with different treatments.
Answer Obviously, the morphology of sample was not affected by soaking time. Therefore, the morphology is shown in the same soaking time of 10 min (figure 2). Moreover, the SEM morphology of 9 QBJR samples has been displayed which showed at the same soaking time of 10 min.
(4) What is the significance of the Fuzzy Analytical Method establishment for this work?
Answer The FAM is a mathematical technique and has been performed for decision. So far, the application of FAM on more than 7 criteria and each criterion contained unequal data has never been reported before. In this work, there are 3 criteria and each criterion consisted of 3 sub-criteria with different weight according to its impact. Moreover, in each QBJR sample, the data of each sub-criteria in physical properties, bioactive compound criteria consisted of 3 experiment data, while the data of sensory score criteria consisted of 30 experiment data. Therefore, with this complex data, the FAM would significance for decision.
(5) The article did not specify the gender or age composition of the sensory evaluation panel. The sensory evaluation data is suggested to be represented in the form of radar maps.
Answer The detailed about the sensory test regarding to panelist gender and age has been added to the material and method part. Moreover, the sensory evaluation data has been represented in the radar map regarding to reviewer suggestion.
(6) Complete the error bars in Fig. 4.
Answer According to each sub-criterion consisted of unequal data, therefore all 108 data of each sample are translated into the fuzzy scale with each weight.
Reviewer 3 Report
The subject of the manuscript is very interesting and with possible practical implementation. It can be considered for publication after some minor corrections.
General Comment:
According to authors all analysis methods are already available in literature sources. Never the less this work is missing information how all methods were tested to ensure proper results are obtained. At least basic method transfer is needed to check specificity, range, repeatability etc. I believe such tests were performed but not described as method development was not subject of this work. Please include information how literature methods performance in this specific use were evaluated. This is especially important for spectrophotometric methods. A summary of these activities should be included.
Specific Comments:
Line 302: (Materials and Methods): Please provide the reagents and chemicals as well as individual standards for the determined polyphenols, flavonoids and anthocyanins, and their source of origin
Line 305: Indicate the applied agronomical practices: irrigation, fertilization, pruning, in addition temperatures during the harvest year (maximum, minimum, mean), the age of the plant.
Lines: 345, 355, 362: Please provide standard curves for individual spectrophotometric methods.
Line 352-354: What are these sentences about?
Line 362: What is the literature on this method?
Line 367: Please check the counting method again and describe it in details.
Author Response
Answer the Reviewer 3
General Comment:
According to authors all analysis methods are already available in literature sources. Never the less this work is missing information how all methods were tested to ensure proper results are obtained. At least basic method transfer is needed to check specificity, range, repeatability etc. I believe such tests were performed but not described as method development was not subject of this work. Please include information how literature methods performance in this specific use were evaluated. This is especially important for spectrophotometric methods. A summary of these activities should be included.
Answer the information /detailed of the analysis method has been added.
Specific Comments:
(1) Line 302: (Materials and Methods): Please provide the reagents and chemicals as well as individual standards for the determined polyphenols, flavonoids and anthocyanins, and their source of origin
Answer the detail of chemical reagent has been added to the material and method part in line 343-351.
(2) Line 305: Indicate the applied agronomical practices: irrigation, fertilization, pruning, in addition temperatures during the harvest year (maximum, minimum, mean), the age of the plant.
Answer the information regarding to the growing of sample such as fertilization, temperature has been added in line 340-342.
(3) Lines: 345, 355, 362: Please provide standard curves for individual spectrophotometric methods.
Answer the standard curve of individual spectrophotometric methods has been added in line 401-403, 406.
(4) Line 352-354: What are these sentences about?
Answer These sentences have been deleted.
(5) Line 362: What is the literature on this method?
Answer The literature on the method has been added (line 410-411)
(6) Line 367: Please check the counting method again and describe it in details.
Answer the counting method has been checked.
Round 2
Reviewer 2 Report
All comments have been addressed and implemented. It is necessary to improve the grammar.
Author Response
This manuscript has been approved by the English lecturer of Arts Program in English for Business Management, Faculty of Interdisciplinary Studies, Khon Kaen University. The English grammar was edited through the manuscript regarding to the reviewer’s comment.
